# Fabrication of Injectable Chitosan-Chondroitin Sulfate Hydrogel Embedding Kartogenin-Loaded Microspheres as an Ultrasound-Triggered Drug Delivery System for Cartilage Tissue Engineering

**DOI:** 10.3390/pharmaceutics13091487

**Published:** 2021-09-16

**Authors:** Fu-Zhen Yuan, Hu-Fei Wang, Jian Guan, Jiang-Nan Fu, Meng Yang, Ji-Ying Zhang, You-Rong Chen, Xing Wang, Jia-Kuo Yu

**Affiliations:** 1Department of Sports Medicine, Peking University Third Hospital, Beijing 100083, China; 1916384020@bjmu.edu.cn (F.-Z.Y.); guanjian@bjmu.edu.cn (J.G.); 1610301219@pku.edu.cn (J.-N.F.); 15650257005@163.com (M.Y.); zjychh@163.com (J.-Y.Z.); chenyourong1990@pku.edu.cn (Y.-R.C.); 2Beijing Key Laboratory of Sports Injuries, Institute of Sports Medicine of Peking University, Beijing 100191, China; 3Beijing National Laboratory for Molecular Sciences, Institute of Chemistry, Chinese Academy of Sciences, Beijing 100190, China; hfwang@iccas.ac.cn; 4University of Chinese Academy of Sciences, Beijing 100049, China

**Keywords:** ultrasound-responsive, microspheres, kartogenin, cartilage tissue engineering

## Abstract

Ultrasound-responsive microspheres (MPs) derived from natural polysaccharides and injectable hydrogels have been widely investigated as a biocompatible, biodegradable, and controllable drug delivery system and cell scaffolds for tissue engineering. In this study, kartogenin (KGN) loaded poly (lactide-*co*-glycolic acid) (PLGA) MPs (MPs@KGN) were fabricated by premix membrane emulsification (PME) method which were sonicated by an ultrasound transducer. Furthermore, carboxymethyl chitosan-oxidized chondroitin sulfate (CMC-OCS) hydrogel were prepared via the Schiff’ base reaction-embedded MPs to produce a CMC-OCS/MPs scaffold. In the current work, morphology, mechanical property, porosity determination, swelling property, in vitro degradation, KGN release from scaffolds, cytotoxicity, and cell bioactivity were investigated. The results showed that MPs presented an obvious collapse after ultrasound treatment. The embedded PLGA MPs could enhance the compressive elastic modulus of soft CMC-OCS hydrogel. The cumulative release KGN from MPs exhibited a slow rate which would display an appropriate collapse after ultrasound, allowing KGN to maintain a continuous concentration for at least 28 days. Moreover, the composite CMC-OCS@MPs scaffolds exhibited faster gelation, lower swelling ratio, and lower in vitro degradation. CCK-8 and LIVE/DEAD staining showed these scaffolds did not influence rabbit bone marrow mesenchymal stem cells (rBMMSCs) proliferation. Then these scaffolds were cultured with rBMMSCs for 2 weeks, and the immunofluorescent staining of collagen II (COL-2) showed that CMC-OCS hydrogel embedded with MPs@KGN (CMC-OCS@MPs@KGN) with ultrasound had the ability to increase the COL-2 synthesis. Overall, due to the improved mechanical property and the ability of sustained KGN release, this injectable hydrogel with ultrasound-responsive property is a promising system for cartilage tissue engineering.

## 1. Introduction

Injuries to articular cartilage are a common health problem, causing pain and disability in all age groups [1]. Due to the lack of vasculature, neural, and lymphatic network and low chondrocyte density, the regenerative capacity of articular cartilage after damage or disease is limited [2]. The field of tissue engineering holds great promise for the cartilage repair and regeneration [3]. Hydrogels are polymeric materials with a synthesizing 3D crosslinked hydrophilic structure [4]. Injectable hydrogel is a unique type of gel for cell scaffolds and drug carries due to their in situ injection, porous structures, and biocompatibility [5]. Wherein, chitosan and chondroitin sulfate-based hydrogels were widely used as drug carriers for cartilage tissue engineering [6,7]. Chitosan is a mucopolysaccharide with a structure similar to glycosamine. It is made up of d-glucosamine and *N*-acetylglucosamine units connected by β(1,4)-glycosidic bonds [8]. As one only positively charged cationic polysaccharide in nature, chitosan has good biocompatibility [9], antibacterial activity [10], and can be slowly degraded within the human body [11], and the degradation products are safe and non-toxic. The chitosan derivative carboxymethyl chitosan is obtained by reacting chitosan with chloroacetic acid under alkaline conditions. It has good water solubility under neutral conditions, and retains the excellent characteristics of chitosan such as biocompatibility, biodegradability, and biological activity [12]. It is one of the most studied chitosan derivatives in recent years. Chondroitin sulfate is produced by the sulfation of chondroitin and is abundant in the extracellular matrix of tissues. Chondroitin sulfate is a glycosaminoglycan composed of β-1,3-linked glucuronic acid (Glca) and (β-l,4) *N*-acetylgalactosamine (GalNAc) alternating units. The location of 4 or 6 position of the galactosamine residue can be sulfated [13]. As one of the main components of the extracellular matrix of chondrocytes, chondroitin sulfate supports cell adhesion, migration [14], proliferation, and differentiation [15], and has an auxiliary effect on tissue formation.

Kartogenin (KGN) is a stable nonprotein small molecule with a structure of 2-[(4-phenylphenyl) carbamoyl] benzoic acid which can induce bone marrow mesenchymal stem cells (BMMSCs) differentiation into chondrocytes by adjusting the CBFβ–RUNX1 signaling pathway [16,17,18]. The effect of KGN on BMMSCs in vitro was dose-dependent and low doses of KGN (10 nmol/L) could effectively induce chondrogenesis [19]. Furthermore, the homing capacity of endogenous host mesenchymal stem cells (MSCs) could be enhanced, which allows the cartilage regeneration without cell transplantation [20]. KGN will be delivered into the circulatory system when KGN itself enters the articular cavities via intra-articular injection [21]. Hence, several drug-delivery systems have been used to achieve prolonged and sustained release of drugs in the joint, such as hydrogels [7], nanoparticles [22], microparticles [23], and liposomes [24]. Poly (lactide-*co*-glycolide) (PLGA) is a biodegradable polymer used in many drug delivery products approved by the U.S. Food and Drug Administration (FDA) due to its biocompatibility. The efficiency of drug release can be changed by varying the composition, molecular weight, and chemical structure of PLGA [25] which has been used in cartilage regeneration as drug carrier [26,27] or cell carrier [28].

Ultrasound can accelerate the polymer degradation and the release of incorporated substances [29], and has been widely studied and applied as one kind of external mechanical force stimuli in drug delivery systems due to its non-invasiveness, high controllability, and high tissue penetration ability [30,31]. However, most of the microbubbles stimulated by ultrasound are synthesized by the liposomes [32,33,34] and the totally collapse of microbubbles always cause the sudden release. Thus, reducing the KGN dose and loss and prolonging the duration of the KGN activity would need to be further studied by a sustained release system.

In the present study, a kind of PLGA microspheres (MPs) effected by ultrasonic condition was fabricated and the injectable chitosan-chondroitin sulfate hydrogel embedded with ultrasound-responsive PLGA MPs encapsulated with KGN (MPs@KGN/ultrasound) was easily synthesized and characterized. We speculated that in terms of controlled drug release, hydrogel had the characteristics of explosive release, while microspheres, nanospheres and liposomes had the characteristics of slow and incomplete release. Thus, the aim of this study was to verify the above-mentioned controlled release system for KGN-sustained release and evaluate the systems for promoting directed chondrogenic differentiation of bone marrow mesenchymal stem cells in vitro.

## 2. Results

### 2.1. Characterizations of PLGA MPs@KGN

The PLGA MPs with and without KGN loading were prepared by the premix membrane emulsification method. They were consistently spherical in shape and exhibited a discrete and smooth surface without any pinholes or cracks. Both MPs with and without KGN displayed an obvious loss after ultrasound treatment for 2 and 5 min in structure showing an evident collapse. After soaking in phosphate buffered saline (PBS) for one month, obvious shrinkage could be seen on the surface of the MPs without cracks (Figure 1A,B). In order to effectively control the KGN release rate, we used ultrasonic treatment for 2 min each time.

As shown in Figure 1C, histograms represented the size distribution of MPs. The diameter of MPs was found to be in the range of 9.2–11.8 μm. Additionally, the medium sizes of MPs with and without KGN were 10.1 μm. In general, owing to their high surface area to volume ratio of the larger size of MPs, the drug loading rate would be increased [35]. The KGN loading content was measured through UV spectrophotometry at 279 nm and the concentration was calculated by the standard curve. The encapsulation efficiency and drug loading capacity of PLGA MPs were (36.02 ± 0.78)% and (0.16 ± 0.05)%, respectively.

Figure 1D showed the KGN release behavior from MPs, which exhibited a low burst-release within the initial 24 h. Once exposed to ultrasound, the KGN could be quickly released with a clear burst-release of KGN each time.

### 2.2. Fabrication and Characterization of Hydrogel Scaffolds

#### 2.2.1. Synthesis of Hydrogel Scaffolds

Figure 2A showed a schematic illustration of the cross-linked reaction in the CMC-OCS hydrogel. CMC could react with oxidized chondroitin sulfate (OC) through a Schiff base reaction to form a stable structure. KGN was encapsulated into PLGA MPs which were mixed in hydrogels. FTIR was used to get further insight into the molecular organization of the polysaccharide derivatives and cross-linked hydrogels (Figure 2B). Compared with chondroitin sulfate (CS), the spectrum of OCS showed an absorption of 1715.3 cm^−1^ which was assigned to the aldehyde groups. The spectrum of CMC-OCS hydrogel was similar to that of CMC in which the peak at 1241.9 cm^−1^ represented the C–O stretching vibrations. In addition, the disappearance of peak at 1715.3 cm^−1^ indicated that OCS was involved in the gelation through Schiff base reaction. The chemical structure of CS, OCS, and CMC are shown in Figure 2C–E. The hydroxyl groups of CS were oxidized into aldehyde groups of OCS using the sodium periodate.

#### 2.2.2. Characterization of Hydrogel Scaffolds and MPs-Incorporated Scaffolds

The hydrogel and hydrogel@MPs were easy to be extruded from a medical syringe (Figure 3A). After mixing with MPs, colorless and transparence hydrogel scaffolds became white and opaque (Figure 3B,C). According to the cross-sectional SEM photographs, both hydrogel and hydrogel@MPs exhibited an interconnected porous 3D framework which enabled 3D culture for cell activity, and the MPs were distributed evenly in the hydrogel to form a composite scaffold with slightly rougher internal structure than the hydrogel scaffold without MPs which might facilitate cell adhesion (Figure 3D). Furthermore, slicing two full hydrogels into halves and reassembling them with each other after exchange could intuitively demonstrate the healing capability of broken hydrogel (Figure 3E). The high adhesiveness of the materials enables instant reconnection. After 15 min of healing, an ambiguous boundary at the healed interface was exhibited by the assembly material, which could maintain the integrity of the healing interface despite subjected to external tensile strength. As a widely used hydrogel material, diverse benefit for a scaffold could be brought by the stable self-healing effect (e.g., injectability, flexible delivery, and adaptability) [36,37,38].

The pore sizes of CMC-OCS, KGN-incorporated CMC-OCS (CMC-OCS@KGN), PLGA MPs@KGN-embedded CMC-OCS (CMC-OCS@MPs@KGN) were 33.03 ± 1.19 µm, 32.71 ± 1.20 µm, and 24.80 ± 1.33 µm, respectively (Figure 4A). Figure 4B shows the porosities of CMC-OCS, CMC-OCS@KGN, CMC-OCS@MPs@KGN as 84.60 ± 5.285%, 82.20 ± 3.43%, and 72.2 ± 3.49%, respectively. The interconnected porous structures can transport nutrients and metabolic waste. The gelation times of CMC-OCS, CMC-OCS@KGN, and CMC-OCS@MPs@KGN were 221.80 ± 12.07 s, 233.60 ± 12.60 s, and 264.60 ± 25.18 s, respectively (Figure 4C). By introducing solid MPs, gelation time was increased significantly. According to the stress–strain curve, the compressive elastic modulus of CMC-OCS and CMC-OCS@KGN hydrogels were 4.90 ± 0.68 kPa and 5.78 ± 0.62 kPa, while hydrogel@MPs showed a significantly larger compressive modulus (23.40 ± 1.34 kPa), which indicated that mixing with MPs could greatly improve the mechanical properties for the cartilage repair (Figure 4D).

The rheological results further demonstrated that the storage modulus (G′) and loss modulus (G″) of CMC-OCS and CMC-OCS@KGN hydrogel scaffolds remained constant within the strain range of 0.01% to 5%, and then it was detected that the value of G′ was lower than G″, indicating that the structure was damaged under large deformations (Figure 4E). Moreover, the structures of CMC-OCS@MPs@KGN hydrogel scaffolds were destroyed under smaller deformations (3%). Furthermore, all the hydrogel scaffolds exhibited similar shear thinning behaviors (Figure 4F) confirming that the hydrogels can be smoothly squeezed out of the syringe. The incorporation of MPs improved the modulus and viscosity of hydrogel scaffolds, resulting in better mechanical properties.

Figure 4G showed that equilibrium swelling ratio of all hydrogel and hydrogel@MPs could be reached in PBS at 37 °C in 96 h. It was noticeable that compared with CMC-OCS and CMC-OCS@KGN hydrogels, hydrogel@MPs exhibited significantly lower swelling ratio, which was consistent with another study (21), since MPs may take up a portion of space of porous hydrogel. The in vitro degradation was monitored as a function of incubation time in PBS at 37 °C. Hydrogel@MPs had an influence on weight loss as shown in Figure 4H. After incubation for 14 days, the degradation ratio of CMC-OCS and CMC-OCS@KGN hydrogels (43.73 ± 6.52% and 44.76 ± 5.91%) was higher than that of hydrogel@MPs (35.42 ± 3.65%). The values of the three groups were 43.73 ± 6.52%, 44.76 ± 5.91%, and 35.42 ± 3.65% after 21 days of culture, which was consistent with swelling and compressive modulus properties. The KGN was loaded into the hydrogel by physical absorption or MPs encapsulation. The cumulative release behavior of KGN in CMC-OCS@KGN was significantly different from that in CMC-OCS@MPs@KGN. In the CMC-OCS@KGN group, KGN showed obvious burst release behavior at the initial release stage, and the burst release rate was 41.83 ± 3.70% and the release rate was 94.35 ± 3.76% at 28 days. In the hydrogel@MPs group, KGN showed lower burst release behavior at the initial release stage. The cumulative release rate of 12 h and 30 days was 20.94 ± 2.64% and 44.51 ± 2.77%, respectively. Upon exposure to ultrasound, KGN showed a burst release from MPs@KGN and CMC-OCS@MPs@KGN. The cumulative release rate of 28 days increased to 74.71 ± 4.03% and 72.52 ± 1.75%, respectively (Figure 4I).

#### 2.2.3. The Cytocompatibility of Hydrogel Scaffolds and Scaffolds Incorporated with MPs In Vitro

Cell compatibility of scaffolds was evaluated by cell viability and cell proliferation by CCK-8 and LIVE/DEAD staining as shown in Figure 5. The results showed that the cell viability ratio of all groups was higher than 90%, which indicated that all kinds of scaffolds and extra ultrasound had good cell compatibility (Figure 5A). No statistical differences were observed among the four groups. The rabbit BMMSCs (rBMMSCs) proliferation on scaffolds was analyzed by CCK-8 (Figure 5B). There was no statistical differences among four groups at 1, 3, 5, and 7 days. When it came to the 10th day, cell proliferation slowed in the KGN release group, and there were statistical differences among the four groups. Since the proliferation rate of chondrocytes was slower than that of rBMMSCs, we speculated that the KGN-released groups promoted the chondrogenic differentiation of rBMMSCs. LIVE/DEAD assay (Figure 5C) was performed after seeding cells onto the scaffolds for 7 days. Live/dead assay showed a large area of living cells (green) and few dead cells (red) in these four groups, indicating good biocompatibility of the scaffold.

#### 2.2.4. Bioactivity of Scaffolds In Vitro

To evaluate the effect of the scaffolds or scaffolds incorporating with ultrasound on differentiation of rBMMSCs in vitro, they were stained with collagen II (COL-2) immunohistochemistry after 14 days. As shown in Figure 6, after 14 days of incubation in vitro, COL-2 gradually increased and COL-2 in CMC-OCS@MPs@KGN/ultrasound group was significantly higher than that of CMC-OCS@MPs@KGN and CMC-OCS@KGN groups (Figure 6B). The CMC-OCS group barely showed any staining for COL-2. The results showed that controlled KGN-released through ultrasound significantly increases the expression of COL-2, which might be helpful for the repair of cartilage.

## 3. Materials and Methods

### 3.1. Materials

Carboxymethyl chitosan (Mw = 100,000, 80% degree of carboxylation, 10–80 mPa·s viscosity) was purchased from Dalian Meilum Biotechnology Co., Ltd. Dalian, China. Chondroitin sulfate (CS, Mn = 19.9 kDa) was obtained from Xingyuan Chemical Reagent Co., Henan, China. Sodium periodate (NaIO_4_) was purchased from SinoPharm Chemical Reagent Co., Ltd. Shanghai, China. PLGA (lactide to glycolide ratio is 70:30, Mw = 10 kDa) and KGN was obtained from Ark Pharm, Inc. Poly (vinyl alcohol) (PVA) (M_w_ = 30,000–70,000, hydrolysis of 87–89%) was purchased from Sigma-Aldrich (St. Louis, MO, USA). All other chemicals and regents were used as received.

### 3.2. Preparation of PLGA MPs@KGN

The PLGA MPs@KGN were prepared via an oil-in-water (O/W) emulsion method combined with premix membrane emulsification. Briefly, PLGA and KGN were dissolved in component solvent of dichloromethane (DCM) and dimethyl sulfoxide (DMSO) (50:1 for volume ratio). Under mechanical stirring, the coarse double emulsion was prepared by pouring the solvent into an aqueous PVA solution immediately. Then, under a certain nitrogen pressure, the crude double emulsion was extruded several times through the SPG membrane. Then, MPs were collected by centrifugation, washed three times with distilled water, and lyophilized.

### 3.3. Surface Morphology and Size Distribution Measurements of PLGA MPs

The MPs and MPs@KGN dispersion were sonicated for 2 or 5 min using an ultrasound transducer (Chattanooga Co., Chattanooga, TN, USA, Thermal index/Ti = 0.9 and Mechanical index/Mi = 1.6) which had been applied in a previous study [39]. Meanwhile, they were soaked in PBS solution (pH = 7.4) at room temperature for one month. The surface morphology and shape of PLGA MPs with or without ultrasound were specified by scanning electron microscope (SEM, FEI Quanta 200, Thermo Fisher Scientific, Waltham, MA, USA). The particle size distribution of MPs used in the experiment was characterized using a laser particle size analyzer (Zetasizer Nano ZS90, Malvern, Malvern, UK).

### 3.4. Measurement of KGN Loading Efficiency and Encapsulation Efficiency

The prepared 5 mg of PLGA MPs@KGN (*n* = 5) were dissolved in component solvent of dichloromethane (DCM) and ethanol (4:1 for volume ratio) of 5.0 mL. The KGN concentration in the solution was spectrophotometrically measured at 279 nm. The drug loading efficiency (LE) and encapsulation efficiency (EE) of the MPs were calculated using the following Equations (1) and (2), respectively.
LE (%) = (Amount of KGN/Mass of PLGA MPs@KGN) × 100%(1)
EE (%) = (Drug loading/Theoretical drug loading) × 100%(2)

### 3.5. KGN Release from Micropheres

For the release study, 5 mg of lyophilized PLGA MPs@KGN (*n* = 5) were placed in PBS solution (pH = 7.4, 3 mL) at 37 °C under gentle agitation. At each predetermined time point (1, 4, 7, 10, 14, 21 and 30 days), for 3 samples, the solution was sonicated for 2 min using an ultrasound transducer. Then we withdraw the supernatants of the 3 samples after being centrifuged for 2 min for further analysis. Subsequently, the MPs were redispersed in the same volume of fresh PBS by vortexing and then incubated until the next release time point.

### 3.6. Synthesis of OCS

OCS was synthesized according to a slightly modified reporting procedure [40]. After dissolving 5 g of CS in 100 mL of distilled water, it was reacted with 3.5 g of NaIO_4_ in the dark at room temperature, and stirring was continued for 6 h. Total of 2.5 mL of ethylene glycol was added to the flask and stirred for 1 h. Distilled water was dialyzed for 3 days to remove residual periodate and ethylene glycol. A white foam material was obtained after the purified product was lyophilized. The actual aldehyde content of OCS was measured by elemental analyzer (Thermo Flash Smart, Milan, Italy), and the oxidation degree was found to be 53.6%.

### 3.7. Fabrication of Hydrogel Scaffolds

To prepare the CMC-OCS hydrogel, CMC (30 mg/mL) and OCS (100 mg/mL) were dissolved in PBS (pH = 7.4) and mixed at a volume ratio of 4 at room temperature. To form MPs-embedded hydrogel, PLGA MPs@KGN was added to OCS solution and OCS@MPs were mixed with CMC solution. MPs were embedded within hydrogel at a concentration of 0.6 mg/mL. The gelation time of hydrogel scaffolds were obtained by vial tilting methods [41].

Fourier transform infrared spectroscopy (FTIR, TENSOR II, Bruker, Billerica, MA, USA) was employed to confirm the chemical structure of CS, OCS, CMC as well as CMC-OCS hydrogels.

### 3.8. Surface Morphology of Hydrogel Scaffolds

The surface morphologies of the PLGA MPs, hydrogel, and hydrogel@MPs were investigated with SEM. Briefly, the frozen MPs, sliced hydrogel, and hydrogel@MPs were gold-coated and observed using SEM at a 3 kV accelerating voltage.

### 3.9. Gelation Time and Self-Healing Performance

The gelation time of different formulations of hydrogels was measured by the vial inversion method [42]. Briefly, 2 mL of hydrogel was added to a glass bottle at 37 °C for different periods of time. When the hydrogel solution no longer flowed after tilting the glass bottle, the gelation time was determined. Two adhesives were fully set in tablet shape and dyed with different colors into halves and reconnected after exchange to characterize the self-healing behavior of the hydrogel. After 15 min of healing, tensile stress was manually applied by reconnecting samples with tweezers.

### 3.10. Porosity Determination

The porosity of hydrogel (*n* = 5) and hydrogel@MPs (*n* = 5) was calculated through ethanol displacement method. Samples were cut into regular cylinders, and then the initial volume and weight (V_0_ and W_0_) were calculated and weighted. Then samples were saturated in ethanol at room temperature for 5 min and reweighted (W_1_) [43]. The porosity of hydrogel could be obtained as follows:Porosity (%) = [(W_1_ − W_0_)/(V_0_ × ρ)] × 100%(3)
ρ represents the density of ethanol.

### 3.11. Rheological Test of Hydrogel Scaffolds

Different rheological characteristics of CMC-OCS, CMC-OCS@KGN, and CMC-OCS@MPs@KGN hydrogel scaffolds were carried out on a rheometer (Thermo Haake Rheometer, Newington, NH, USA). For cyclic alternating shear strain sweep tests, with T = 30 °C, *f* = 0.02 Hz, the shear strain was changed successively from strain = 0.01% to strain = 6% (60 min), and the cycle was repeated for three times, and was tested under the action of 6% strain. After each phase of the test, the next phase was immediately switched without retention. At least three effective samples were tested for each hydrogel sample. For shear rate sweep tests, shear viscosity was recorded over a shear rate range of 0.004 to 0.4 s^−1^.

### 3.12. Swelling Properties

The dry hydrogel (*n* = 5) and hydrogel@MPs (*n* = 5) of the same size weighted W*_d_* were incubated in phosphate-buffer saline (PBS, pH = 7.4) at 37 °C for certain times (0.25, 0.5, 1, 2, 4, 6, 12, 24, 48, 72, 96, 120, 144, and 168 h). The samples were taken out from swelling medium and placed between two sheets of paper to remove the excess water on surface and weighted W*_s_*. The swelling ratio was calculated as below:Swelling ratio (%) = (W*_s_*−W*_d_*)/W*_d_* × 100%(4)

### 3.13. Compressive Testing

The compressive testing of hydrogel (*n* = 5) and hydrogel@MPs (*n* = 5) were calculated by a universal material testing machine of Instron 3365 (Instron Co., Norwood, MA, USA). Samples were cut into cylinders (diameter 15 mm and height 7.5 mm) and the compressive testing was carried out with a beam velocity of 3 mm/min. The compressive elastic modulus was calculated according to the stress–strain curve [44].

### 3.14. KGN Release Profile from Scaffolds

To examine the KGN release profile, hydrogel@MPs (5.0 mg, *n* = 5) were placed in PBS solution (pH = 7.4, 1.0 mL) and agitated continuously at 37 °C. At specific time intervals, PBS solution was obtained and an equivalent amount of fresh PBS was added. The concentration of KGN was measured by UV spectrophotometry at 279 nm.

### 3.15. In Vitro Degradation

The degradation of hydrogel and hydrogel@MPs (*n* = 5) in vitro was examined through weight loss in PBS solution during 21 days. Initial weight of lyophilized samples (W_0_) were recorded and immersed in PBS solution at 37 °C for certain time (1, 4, 7, 14, and 21 days). Then the samples were taken out, lyophilized and weighted again (W_1_). The degradation ratio was defined as follows:Degradation ratio (%) = [(W_0_ − W_1_)/W_0_] × 100%(5)

### 3.16. In Vitro Studies of Scaffolds

#### 3.16.1. In Vitro Cytotoxicity

Cytotoxicity of hydrogel was measured using Cell Counting Kit-8 (CCK-8, Dojindo, Kumamoto, Japan) assay system at 12, 24, and 48 h post-treatment by contacting the extracts of hydrogel. The rBMMSCs (P2) (2 × 10^3^/100 μL/well) were seeded in 96-well microplates and then incubated at 37 °C in 5% CO_2_ for 12 h to obtain a monolayer of cells. Then hydrogel extracts (*n* = 3) or PBS (10 μL) were added to each well and incubated for predetermined time. After 12, 24, and 48  h of incubation, the cell culture medium was removed and then 100 μL of fresh culture medium and 10  μL of CCK-8 were added to the 96 wells for 2 h. Subsequently, 110 μL was transferred to a 96-well plate, and the absorbance was read at 450 nm on a microplate reader (Thermo, Waltham, MA, USA). Cell viability (%) was calculated using the following Equation:Cell viability (%) = [(Asample − Ablank)/(Acontrol − Ablank)]  ×  100%(6)

The data represented the mean of six independent experiments and were expressed as means ± SD. Cell viability of <70% was considered cytotoxic [41].

#### 3.16.2. Cell Proliferation on Scaffolds

Cell proliferation was measured with the CCK-8 assay kit. The rBMMSCs (P2) were cultured with hydrogel and hydrogel@MPs (*n* = 3) for 1, 3, 5, 7, and 10 days. When treatments were completed, the cell culture medium was substituted with 100 μL fresh cell culture medium and 10 μL of CCK-8, and then the cultures were incubated for 2 h at 37 °C. The absorbance (OD) value at 450 nm for each well was recorded by using a microplate reader. Cell proliferation curves were depicted according to the OD value at indicated time points.

#### 3.16.3. Cell Seeding on Scaffolds

Total of 50 μL concentrated rBMMSCs solution (1.0 × 10^7^ cells/mL) was seeded in lyophilized hydrogel and hydrogel@MPs (*n* = 3) (2 mm in thickness and 6 mm in diameter) by centrifugation [45]. After cell implantation, the scaffold was incubated for 1 h to attach cells, and then 2 mL of fresh culture medium was added.

#### 3.16.4. Cell Compatibility and Distribution on Scaffolds

Cell culture medium was replaced every 3 days and subsequent to being cultured for 7 days, each scaffold was washed in PBS twice, and immersed in 500  μL of PBS with 2  mM of calcein AM and 4  mM of PI reagents (Invitrogen, Carlsbad, CA, USA) before incubation for 1  h at 37 °C. Confocal laser-scanning microscopy (Leica, Wetzlar, Germany) with excitation wavelength of 488  and 568  nm was used to detect live (green) and dead (red) cells.

#### 3.16.5. Cell Bioactivity Assessment of KGN Released from Scaffolds

The cell-seeded scaffolds were fed with chondrogenic differentiation solution (Cyagen Biosciences Inc., Suzhou, China) and the solution was changed every other day. Immunofluorescent staining was used to analyze the formation of COL-2 in cartilage scaffold. After being cultured for 14 days, the scaffolds were washed with PBS and fixed with 4% paraformaldehyde at room temperature for 10 min. Then, scaffolds were blocked with 5% bovine serum albumin (BSA) for 30 min before exposure to rabbit anti-COL II primary antibody (1:200, Abcam, Cambridge, UK) overnight at 4 °C followed by secondary antibodies (anti-rabbit IgG, green, Abcam, Cambridge, UK) at 37 °C for 40 min. Then the immunofluorescence images were obtained by using confocal microscopy (Leica TCS SP8, Leica Microsystems Inc., Heidelberg, Germany). By calculating and normalizing the fluorescence intensity of COL-2 to the number of the cells where the COL-2 fluorescence was unevenly distributed in the immunofluorescence images, the amount of COL-2 was quantitatively evaluated by Image-Pro Plus software (6.0; Media Cybernetics Inc., San Diego, CA, USA) (*n* = 3).

### 3.17. Statistical Analysis

All data are presented as the mean ± standard deviation (mean ± SD). Statistical analysis among groups were calculated using a one-way analysis of variance ANOVA after testing for homogeneity of variance (test Levene’s) with post hoc contrasts by least significant difference test (LSD-t). Significance thresholds were set at *p* < 0.05 (significant) and *p* < 0.01 (highly significant). Data analysis was performed using SPSS 25.0 for Windows (SPSS Inc., Chicago, IL, USA).

## 4. Conclusions

In this study, an injectable CMC-OCS hydrogel containing MPs was developed via the Schiff base cross-linking reaction and PLGA MPs@KGN were prepared through the PME method. The obtained CMC-OCS@MPs@KGN exhibited a shorter gelation time and slower weight loss than control hydrogels in vitro. Compressive elastic modulus of the hydrogel@MPs was significantly improved and beneficial to the cartilage repair. Meanwhile, the PLGA MPs@KGN could respond to ultrasound with controlled burst-release of KGN. In vitro rBMMSCs seeded in the scaffolds actually showed significant higher viability during culture time of 7 to 10 days. These results indicated the advanced strategy on fabrication of controlled KGN release which could be used in tissue-engineered cartilage.

## Figures and Tables

**Figure 1 pharmaceutics-13-01487-f001:**
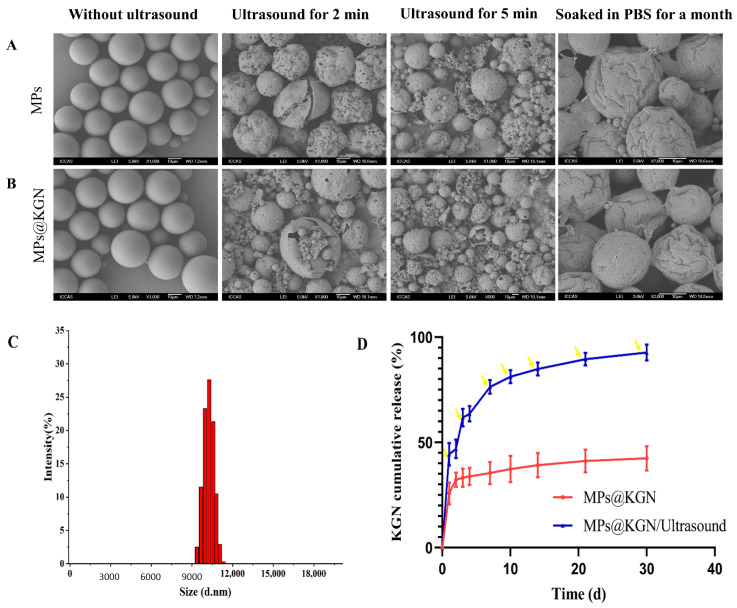
SEM images of MPs with and without KGN: (**A**) MPs without KGN; (**B**) MPs with KGN; (**C**) size distribution; (**D**) in vitro KGN release profiles of PLGA MPs and the yellow arrow represents ultrasound for 2 min.

**Figure 2 pharmaceutics-13-01487-f002:**
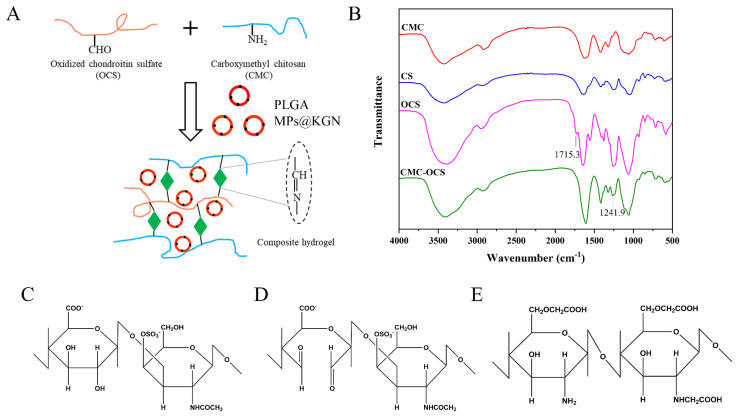
(**A**) Schematic illustration of the cross-linked reaction. (**B**) FTIR spectra of polysaccharide derivatives and cross-linked hydrogels. (**C**–**E**) Chemical structures of CS, OCS, and CMC.

**Figure 3 pharmaceutics-13-01487-f003:**
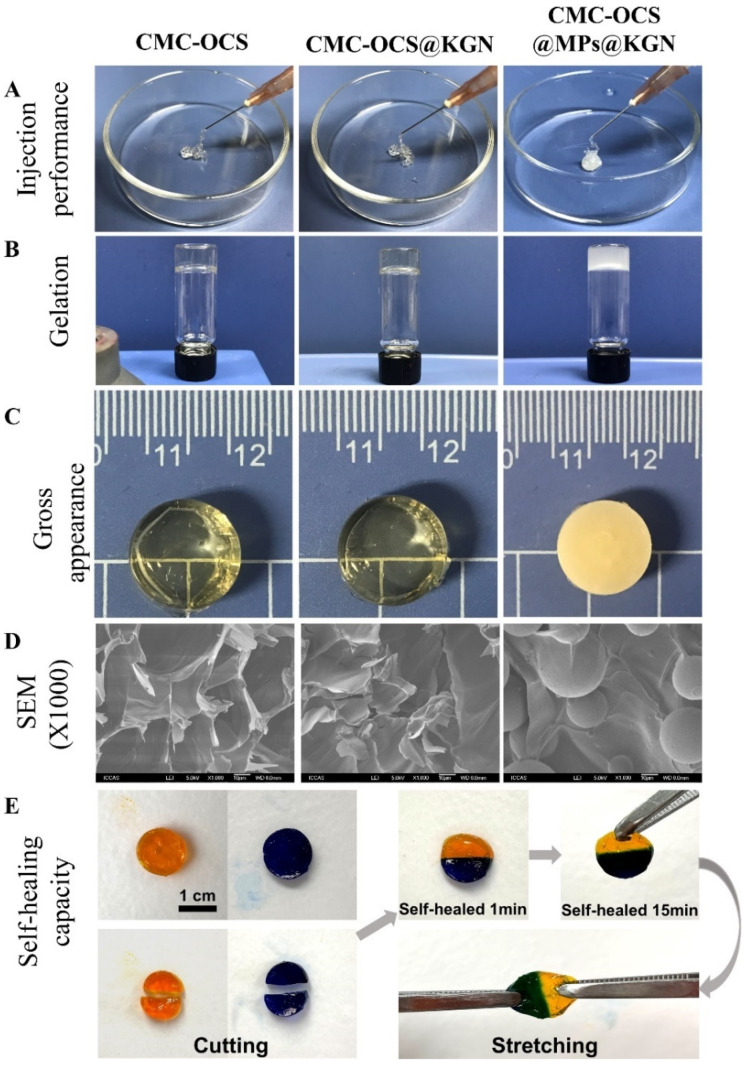
(**A**) The extrusion from a conventional medical syringe showing the injection performance of hydrogel scaffolds. (**B**) Gelation process of scaffolds. (**C**) Gross performance of scaffolds. (**D**) Cross-sectional SEM photographs showing the interconnected porous structure of scaffolds. (**E**) Demonstration on self-healing capacity by cutting hydrogel into halves and rejoining them with each other to achieved healed interfaces.

**Figure 4 pharmaceutics-13-01487-f004:**
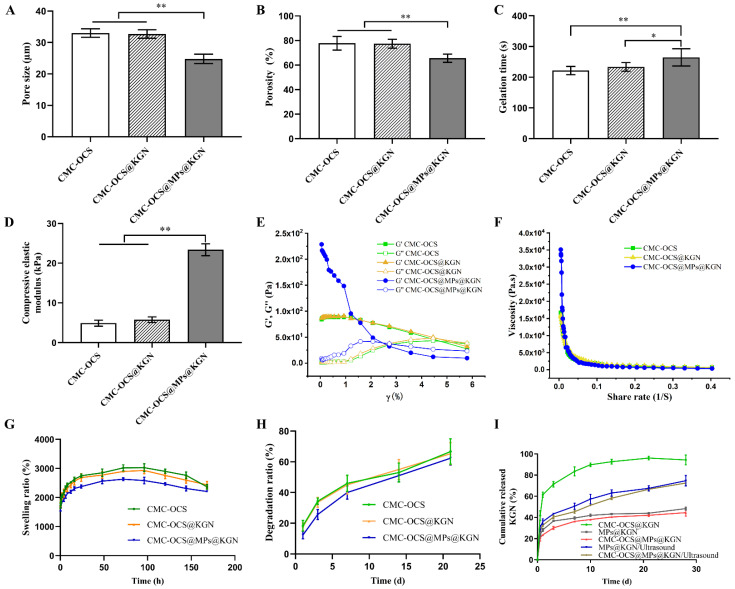
(**A**) The average pore size of three scaffolds (*n* = 5, ** *p* < 0.01). (**B**) Porosity of three scaffolds (*n* = 5, ** *p* < 0.01). (**C**) Gelation time of three scaffolds (*n* = 5, ** *p* < 0.01, * *p* < 0.05). (**D**) Compressive elastic modulus of three scaffolds (*n* = 5, ** *p* < 0.01). (**E**) Storage modulus (G′) and loss modulus (G″) of different hydrogel scaffolds as a function of strain (γ). (**F**) Viscosity measurement of different hydrogel scaffolds in a shear rate sweep from 0.004 to 0.4 s^−1^, indicative of shear-thinning behavior. (**G**) Swelling ratio of three scaffolds at each time point (*n* = 5). (**H**) Degradation ratio of three scaffolds at different time points. (**I**) Cumulative KGN release of MPs or scaffolds with or without ultrasound.

**Figure 5 pharmaceutics-13-01487-f005:**
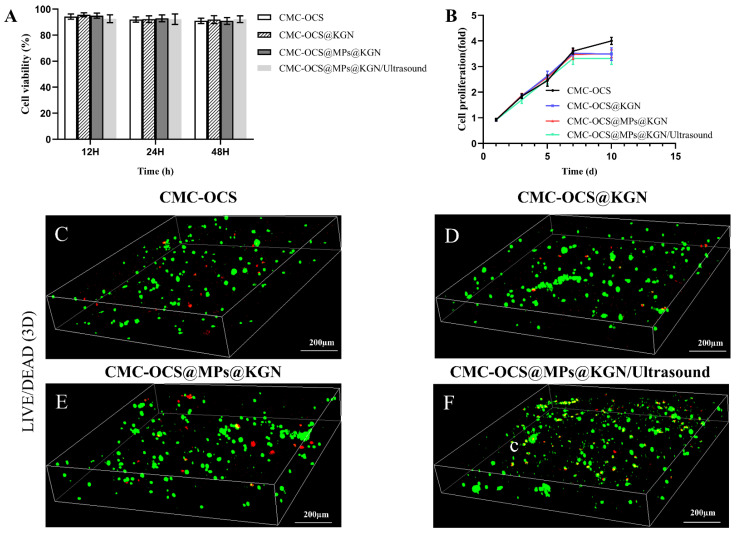
(**A**) Cytotoxicity assay using a CCK-8 kit. (**B**) Cell proliferation in four groups as detected with a CCK-8 assay. Data were normalized against the OD value on day 1 of each cell line. (**C**–**F**) Live/dead staining of rBMMSCs performed after 7 days of incubation with the four groups.

**Figure 6 pharmaceutics-13-01487-f006:**
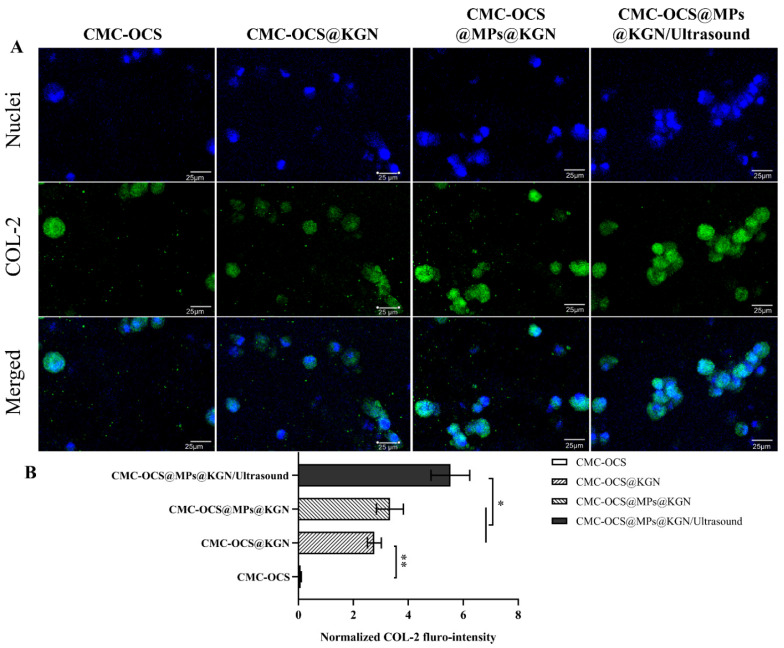
Differentiation status of rBMMSCs induced by chondrogenesis after 14 days of in vitro incubation. Immunofluorescent staining of COL-2 (**A**) and normalized COL-2 fluro-intensity (**B**) in the four groups (** *p* < 0.01, * *p* < 0.05).

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
