# Peer review of "Fabrication of Injectable Chitosan-Chondroitin Sulfate Hydrogel Embedding Kartogenin-Loaded Microspheres as an Ultrasound-Triggered Drug Delivery System for Cartilage Tissue Engineering"

_pharmaceutics, 2021, doi:10.3390/pharmaceutics13091487_

Round 1
Reviewer 1 Report
Chondroitin sulfate-based hydrogels are in fact promising scaffold materials for cartilage tissue engineering. The incorporation of ultrasound-responsive microspheres releasing a cell-stimulating bioactive molecule in such a system represents an interesting approach to improve the application properties of the scaffold system. Therefore, this work has some potential. However, prior acceptance, the manuscript needs some improvements.
Introduction
Line 49: Chondroitin sulfate is a glycosaminoglycan composed of glucuronic acid (GlcA) and N-acetylgalactosamine (GalNAc), not as written, N-acetylglucosamine. Please correct!
Line 52: Kartogenin (KGN) was used to induce the chondrogenesis. A few more words about molecule structure, expected release profile, necessary concentrations would be helpful for the reader.
Results
Figure 2: The illustrations showing chondroitin sulfate (CS) and oxidized chondroitin sulfate (OCS) are not correct. These drawings show sulfonate (SO3), but not sulfate (OSO3) groups as in the case of chondroitin sulfate. Please correct!
Line 112: Concerning the mentioned self-healing effect, references should be included.
Materials and Methods
Line 176: The degree of carboxylation of carboxymethyl chitosan is given as 80%.Does this mean that 80 % of the NH2-groups are carboxymethylated (in Figure 2, only the NH2-groups are carboxymethylated, but in principle, both the NH2- and the OH-groups of chitosan can be carboxymethylated dependant on the reaction conditions. What was the case here? Please specify! Please also give the molecular weight of used carboxymethyl chitosan.
Line 184: PVA is poly (vinyl alcohol)?
Fabrication of hydrogel scaffolds
Line 207: The oxidation of polysaccharides and even glycosaminoglycans with sodium periodate does not result in a complete conversion to dialdehydes (as schematically shown in Fig. 2). Therefore, the degree of oxidation has to be determined (e. g. by using a suitable analytical method). Please comment!
Author Response
Thank you very much for your kind advice and valuable comments in helping us improve our manuscript. We have substantially modified the manuscript, according to the questions raised by the Editor and Reviewers. All the modified words, sentences and paragraphs were labeled with red fonts. A point-to-point response to highlight how we have addressed each of the comments is listed below.
Comments and Suggestions for Authors: Chondroitin sulfate-based hydrogels are in fact promising scaffold materials for cartilage tissue engineering. The incorporation of ultrasound-responsive microspheres releasing a cell-stimulating bioactive molecule in such a system represents an interesting approach to improve the application properties of the scaffold system. Therefore, this work has some potential. However, prior acceptance, the manuscript needs some improvements.
Introduction
Line 49: Chondroitin sulfate is a glycosaminoglycan composed of glucuronic acid (GlcA) and N-acetylgalactosamine (GalNAc), not as written, N-acetylglucosamine. Please correct!
Response 1: We apologize for our carelessness. Thank you for your thoughtful suggestion. We have corrected it (Line 54). “Chondroitin sulfate is a glycosaminoglycan composed of β-1,3-linked glucuronic acid (Glca) and (β-1,4) N-acetyl-galactosamine (GalNac) alternating units.”
Line 52: Kartogenin (KGN) was used to induce the chondrogenesis. A few more words about molecule structure, expected release profile, necessary concentrations would be helpful for the reader.
Response 2: Thank you for your thoughtful suggestion. We have already described KGN in more detail. (Line 59-64, 73-74). “Kartogenin (KGN) is a stable nonprotein small molecule with a structure of 2-[(4-phenylphenyl) carbamoyl] benzoic acid which can induce bone marrow mesenchymal stem cells (BMMSCs) differentiation into chondrocytes by adjusting the CBFβ–RUNX1 signaling pathway [16-18]. And the effect of KGN on BMMSCs in vitro was dose-dependent and low doses of KGN (10 nmol/L) could effectively induce chondrogenesis [19]. Furthermore, it can recruit endogenous host mesenchymal stem cells (MSCs) for homing, thus achieving cartilage regeneration without cell transplantation [20]. Despite the fact that KGN can be delivered into articular cavities via intraarticular injection, most of the KGN will be absorbed by the circulatory system [21].” “Therefore, a sustained release system is necessary to reduce the KGN dose and loss, prolong the duration of the KGN activity, and match with the repair cycle required for damaged cartilage.”
Results
Figure 2: The illustrations showing chondroitin sulfate (CS) and oxidized chondroitin sulfate (OCS) are not correct. These drawings show sulfonate (SO3), but not sulfate (OSO3) groups as in the case of chondroitin sulfate. Please correct!
Response 3: We apologize for our carelessness. We have redrawn the structural formula of chondroitin sulfate and oxidized chondroitin sulfate. (Figure 2C and 2D)
Line 112: Concerning the mentioned self-healing effect, references should be included.
Response 4: Thank you for your thoughtful suggestion. We have added three references in the part 2.2.2. (Line 139).
Materials and Methods
Line 176: The degree of carboxylation of carboxymethyl chitosan is given as 80%. Does this mean that 80 % of the NH2-groups are carboxymethylated (in Figure 2, only the NH2-groups are carboxymethylated, but in principle, both the NH2- and the OH-groups of chitosan can be carboxymethylated dependant on the reaction conditions. What was the case here? Please specify! Please also give the molecular weight of used carboxymethyl chitosan.
Response 5: Yes, your opinion is very rigorous. We only drew the structural formula for the convenience of readers, without considering the degree of substitution of chitosan. We have redrawn it in Figure 2E.
Line 184: PVA is poly (vinyl alcohol)?
Response 6: We apologize for our carelessness. PVA is poly (vinyl alcohol). We have added PVA properties and companies in the part 3.1. Materials. (Line 227-228). “Poly (vinyl alcohol) (PVA) (Mw = 30,000–70,000, hydrolysis of 87–89%) was purchased from Sigma-Aldrich (St. Louis, MO).” Fabrication of hydrogel scaffolds
Line 207: The oxidation of polysaccharides and even glycosaminoglycans with sodium periodate does not result in a complete conversion to dialdehydes (as schematically shown in Fig. 2). Therefore, the degree of oxidation has to be determined (e. g. by using a suitable analytical method). Please comment!
Response 7: Thank you for your thoughtful suggestion. It was a mistake in our writing. We missed the test of OCS oxidation degree in part 3.6. (Line 257-263). “OCS was synthesized according to a reported procedure slightly modified. 5 g of CS was dissolved in 100 mL of distilled water and reacted with 3.5 g of NaIO4 at room temperature in the dark with constant stirring for 6 h. 3.5 mL of ethylene glycol was added to the flask with stirring for 1 h. The dialysis was performed against distilled water for 3 days to remove residual periodate and ethylene glycol. The purified product was lyophilized to obtain a white foam material. Determination of the actual aldehyde content of OCS revealed an extent of oxidation of 53.6% by an element analyzer (Thermo Flash Smart, Italy).”

Reviewer 2 Report
-The author used probe type sonication for triggered release, which is used for the homogenizing tissue. Typical use of ultrasound in medical condition is very different from that of current study. They have to write the information in introduction well.
-How amount of NaIO4 was added to produce oxidized CS as typical experiment? How do you or did you remove NaIO4?
-More information about sonication should be added.
-The effect of sonication on the cytotoxicity should be investigated well.
-There is a lag time to release drug after ultrasound. They should write the explanation.
-The addition of references about PLGA application to articular cartilage (e.g. knee and other tissue) may be useful.
-In practical condition, how do you apply probe type sonication into articular cartilage in body?
Author Response
Thank you very much for your kind advice and valuable comments in helping us improve our manuscript. We have substantially modified the manuscript, according to the questions raised by the Editor and Reviewers. All the modified words, sentences and paragraphs were labeled with red fonts. A point-to-point response to highlight how we have addressed each of the comments is listed below.
Comments and Suggestions for Authors
-The author used probe type sonication for triggered release, which is used for the homogenizing tissue. Typical use of ultrasound in medical condition is very different from that of current study. They have to write the information in introduction well.
Response 1: We apologize for our carelessness and thank you for your thoughtful suggestion. This ultrasound used in current study is the ultrasound in medical application. We think the word probe is ambiguous, so we changed probe to transducer throughout the text. And we added more information including a reference in introduction. (Line 70-72)
“Ultrasound can accelerate polymer degradation and the release of incorporated substances [29], and has been widely studied and applied as one kind of external mechanical force stimuli in drug delivery systems due to its non-invasiveness, high controllability and high tissue penetration ability [30,31]”
-How amount of NaIO4 was added to produce oxidized CS as typical experiment? How do you or did you remove NaIO4?
Response 2: We apologize for our carelessness and we have added this information in the part 3.6. (Line260-261)
“The dialysis was performed against distilled water for 3 days to remove residual periodate and ethylene glycol.”
-More information about sonication should be added.
Response 3: We apologize for our carelessness and thank you for your thoughtful suggestion. The ultrasound instrument we use is mainly used in physiotherapy. The application of this instrument has been published in the ACS nano recently, and we have cited it in the article. At the same time, we have further clarified the model of the instrument. (Line 238-240)
“The MPs and MPs@KGN dispersion were sonicated for 2 or 5 min using an ultrasound transducer (Chattanooga Co., USA, Thermal index/Ti = 0.9 and Mechanical index/Mi = 1.6) which had been applied in a previous study [39].”
-The effect of sonication on the cytotoxicity should be investigated well.
Response 4: Thank you for your thoughtful suggestion. At present, the methods commonly used to detect cytotoxicity are MTT, CCK-8 and LIVE/DEAD cell staining. The MTT effect is equivalent to CCK-8. Therefore, we chose the more commonly used CCK-8 and LIVE/DEAD cell staining to verify the cytotoxicity of materials and ultrasound. And we think these can explain the problem to a certain extent.
-There is a lag time to release drug after ultrasound. They should write the explanation.
Response 5: Thank you for your thoughtful suggestion. As shown in Figure 1D, after the ultrasound, we immediately did the test and found that there was an obvious KGN release. Therefore, we think this lag time is negligible.
-The addition of references about PLGA application to articular cartilage (e.g. knee and other tissue) may be useful.
Response 6: We apologize for our carelessness and thank you for your thoughtful suggestion. We added three references to confirm that PLGA is widely used in cartilage regeneration (line 67-69).
“The efficiency of drug release can be changed by varying the composition, molecular weight and chemical structure of PLGA [25] which has been used in cartilage regeneration as drug carrier [26,27] or cell carrier [28].”
-In practical condition, how do you apply probe type sonication into articular cartilage in body?
Response 7: Thank you for your thoughtful suggestion. For example, when knee osteoarthritis is treated by injecting microsphere sustained-release system, push the patella to one side and place the ultrasonic transducer on the other side of the patella for drug-controlled release according to the concentration and drug release curve.

Reviewer 3 Report
The article needs major modification as there are several issues in the current article. The methodology is not very clear. Authors should explain the results and discussion in more detail with a clear rationale.
Author Response
Comments and Suggestions for Authors
The article needs major modification as there are several issues in the current article. The methodology is not very clear. Authors should explain the results and discussion in more detail with a clear rationale.
Response: Thank you so much for your thoughtful suggestion. We have made the modifications and improvements to the greatest extent based on the opinions of you and other reviewers.
Reviewer 4 Report
KGN was encapsulated in PLGA MPs and the ultrasound was applied for accelerated KGN release. MPs@KGN were then embedded in CMC-OCS hydrogel and chondrogenesis of rBMMSCs was evaluated in vitro. The formulations concept seems to be worked in in vitro model, however none in vivo data were included in this manuscript. Current format seems to be insufficient to be published in this journal. Please see the following points to improve the quality of your manuscript.
1) In Figure 1, degradation profiles of MPs and MPs@KGN without ultrasound treatment observed by FE-SEM imaging and particle size analysis for 30 days (as shown in Figure 1D) should be provided.
2) In Figure 4, more rheological data (strain/frequency sweep and viscosity data) should be included to compare the rheological properties of CMC-OCS, CMC-OCS@KGN, and CMC-OCS@MPs@KGN.
3) In Figure 6, FL intensity data in restricted area are not able to be used for statistical analysis. Also, the expression levels of other markers for chondrogenesis should be provided.
4) Related in vivo data should be included in this manuscript to confirm the usefulness of designed scaffold.
Author Response
Thank you very much for your kind advice and valuable comments in helping us improve our manuscript. We have substantially modified the manuscript, according to the questions raised by the Editor and Reviewers. All the modified words, sentences and paragraphs were labeled with red fonts. A point-to-point response to highlight how we have addressed each of the comments is listed below.
Comments and Suggestions for Authors
KGN was encapsulated in PLGA MPs and the ultrasound was applied for accelerated KGN release. MPs@KGN were then embedded in CMC-OCS hydrogel and chondrogenesis of rBMMSCs was evaluated in vitro. The formulations concept seems to be worked in in vitro model, however none in vivo data were included in this manuscript. Current format seems to be insufficient to be published in this journal. Please see the following points to improve the quality of your manuscript.
1) In Figure 1, degradation profiles of MPs and MPs@KGN without ultrasound treatment observed by FE-SEM imaging and particle size analysis for 30 days (as shown in Figure 1D) should be provided.
Response 1: Yes, your opinion is very rigorous and thanks for your thoughtful suggestion. We have added this information in part 2.1 and 3.3. (Line 94-95, 240).
“After soaking in phosphate buffered saline (PBS) for one month, obvious shrinkage could be seen on the surface of the MPs without cracks (Figure 1A and 1B)”
“Meanwhile, they were soaked in PBS solution (pH = 7.4) at room temperature for one month.”
2) In Figure 4, more rheological data (strain/frequency sweep and viscosity data) should be included to compare the rheological properties of CMC-OCS, CMC-OCS@KGN, and CMC-OCS@MPs@KGN.
Response 2: Yes, your opinion is very rigorous. Thanks for your thoughtful suggestion. We have added rheological data in the revised Figure 4E and 4F (Line 155-161, 183-185 and 300-306).
“Rheological results further showed the storage modulus (G’) and loss modulus (G’’) of CMC-OCS and CMC-OCS@KGN hydrogel scaffolds kept in a constant with the strain from 0.01% to 5%, and then lower value of G’ than G’’ was detected, suggesting destruction of structures under large deformations (Figure 4E). And the structures of CMC- OCS@MPs@KGN hydrogel scaffolds were destroyed under smaller deformations (3%). Furthermore, all the hydrogel scaffolds exhibit similar shear thinning behaviors (Figure 4F) confirming that the hydrogels can be smoothly squeezed out of the syringe. The incorporation of MPs improved the modulus and viscosity of hydrogel scaffolds, resulting in better mechanical properties. “
“3.11. Rheological test of hydrogel scaffolds
Different rheological characteristics of CMC-OCS, CMC-OCS@KGN and CMC-OCS@MPs@KGN hydrogel scaffolds were carried out on a rheometer (Thermo Haake Rheometer, USA). For cyclic alternating shear strain sweep tests, with T = 30 °C, f = 0.02 Hz, the shear strain was changed successively from strain= 0.01% to strain= 6% (60 min), and the cycle was repeated for three times, and was tested under the action of 6% strain. After each phase of the test, the next phase was immediately switched without retention. At least three effective samples were tested for each hydrogel sample. For shear rate sweep tests, shear viscosity was recorded over a shear rate range of 0.004 to 0.4 s-1.”
3) In Figure 6, FL intensity data in restricted area are not able to be used for statistical analysis. Also, the expression levels of other markers for chondrogenesis should be provided.
Response 3: For FL intensity data, we apologize for our mistakes in expression. Then the COL-2 immunofluorescence images were obtained by using confocal microscopy. In the areas where the intensity of COL-2 fluorescence was not evenly distributed, it was not suitable to be analyzed directly. So, we used Image-Pro Plus software to assess the amount of COL-2 quantitatively by calculating and normalizing the fluorescence intensity of COL-2 to the number of the cells in the immunofluorescence images in the revised Figure 6B. (Line 215-216 and 370-373).
“As shown in Figure 6, after 14 days of incubation in vitro, COL-2 gradually increased and COL-2 in CMC-OCS@MPs@KGN/ultrasound group was significantly higher than that of CMC-OCS@MPs@KGN and CMC-OCS@KGN groups (Figure 6B). Barely staining for COL-2 was observed in CMC-OCS group. The results showed that controlled KGN-released through ultrasound significantly increases the expression of collagen- 2, which might be helpful for the repair of cartilage.”
“We used Image-Pro Plus software (6.0; Media Cybernetics) to assess the amount of COL-2 quantitatively by calculating and normalizing the fluorescence intensity of COL-2 to the number of the cells where COL-2 fluorescence was not evenly distributed in the immunofluorescence images (n = 3). “
For other markers for chondrogenesis, your opinion is very rigorous. and thanks for your thoughtful suggestion. KGN is currently mainly used for cartilage repair of osteochondral defects and osteoarthritis. The main component of hyaline cartilage is COL-2, so we choose the marker of COL-2. In the future, in more in-depth research, we will gradually add more markers to verify its application in structures such as meniscus and tendons.
4) Related in vivo data should be included in this manuscript to confirm the usefulness of designed scaffold.
Response 4: Yes, your opinion is very rigorous. Thanks for your thoughtful suggestion. We initially explored the injectable chitosan-chondroitin sulfate hydrogel embedding kartogenin loaded microspheres as an ultrasound-triggered drug delivery system and achieved certain results. We will further verify this research in vivo in the future.

Reviewer 5 Report
The authors present their studies on the fabrication and characterization of kartogenin loaded PLGA microspheres embedded in CMC-OCS hydrogels. It is intended that the hydrogel system be applied as a drug delivery system for cartilage tissue engineering. The paper falls within the scope of the journal but the following should be attended to before it can be considered for publication:
- Title: The title as it is suggests that ultrasound responsiveness is a major feature of the hydrogel system. However, ultrasound responsiveness is only used to distinguish the sustained performance of the drug delivery system in the paper. Therefore, it is imperative that the title be changed to reflect the whole study. A suggestion is " Fabrication of injectable chitosan-chondroitin sulfate hydrogel embedding kartogenin loaded microspheres as a drug delivery system"
- Studies on drug loaded microspheres embedded in in hydrogels exist in literature. What is novel in this study? This should be explained in the introduction section.
- Line 49 is confusing as it is so it should be rewritten. Are the authors implying that both Glca and GalNAc units could be sulfated?
- Line 183: please specify the solvent composition ratios (DCM:DMSO).
- Please specify the source and characteristics of PVA in the materials section.
- Line 194: please specify the solvent composition ratio DCM:ethanol
- Line 189: The "MPs solution" is actually a dispersion. Please correct this.
- Considering the drug release profiles, is it possible to achieve therapeutic levels of drug release without the application of ultrasound?
- It is encouraging to see that post hoc statistical analysis have been applied to the results following ANOVA. Please describe these post hoc analysis in the statistical analysis section.
Author Response
Comments and Suggestions for Authors
The authors present their studies on the fabrication and characterization of kartogenin loaded PLGA microspheres embedded in CMC-OCS hydrogels. It is intended that the hydrogel system be applied as a drug delivery system for cartilage tissue engineering. The paper falls within the scope of the journal but the following should be attended to before it can be considered for publication:
- Title: The title as it is suggests that ultrasound responsiveness is a major feature of the hydrogel system. However, ultrasound responsiveness is only used to distinguish the sustained performance of the drug delivery system in the paper. Therefore, it is imperative that the title be changed to reflect the whole study. A suggestion is " Fabrication of injectable chitosan-chondroitin sulfate hydrogel embedding kartogenin loaded microspheres as a drug delivery system"
Response 1: Thank you for your thoughtful suggestion. According to your suggestion, we decided to change the title to “Fabrication of injectable chitosan-chondroitin sulfate hydrogel embedding kartogenin loaded microspheres as an ultrasound-triggered drug delivery system for cartilage tissue engineering”.
- Studies on drug loaded microspheres embedded in in hydrogels exist in literature. What is novel in this study? This should be explained in the introduction section.
Response 2: Yes, your opinion is very rigorous. In the past, many studies have shown the effectiveness of microspheres loaded with drugs which are affected by many factors. In order to maintain a certain concentration of the drug for a certain period of time, more drugs need to be embedded, and the toxic side effects of drug burst release may occur. This study demonstrated the controlled release effect of ultrasound on microspheres, which can be released on demand, which is relatively innovative. Meanwhile this research provides a new method for cartilage repair and other application fields through the basic research of this stimulus-responsive drug-controlled release system. And we added more information in the part of introduction.
- Line 49 is confusing as it is so it should be rewritten. Are the authors implying that both Glca and GalNAc units could be sulfated?
Response 3: We apologize for our carelessness. Thanks for your thoughtful suggestion. We have made changes to the expression. (Line 54).
“The location of 4 or 6 position of the galactosamine residue can be sulfated”
- Line 183: please specify the solvent composition ratios (DCM:DMSO).
Response 4: We apologize for our carelessness. Thanks for your thoughtful suggestion. We have made a further expression. (Line 232).
“Briefly, PLGA and KGN were dissolved in component solvent of dichloromethane (DCM) and dimethyl sulfoxide (DMSO) (50:1 for volume ratio).”
- Please specify the source and characteristics of PVA in the materials section.
Response 5: We apologize for our carelessness. Thanks for your thoughtful suggestion. We have made a further expression. (Line 277-278).
“Poly (vinyl alcohol) (PVA) (Mw = 30,000–70,000, hydrolysis of 87–89%) was purchased from Sigma-Aldrich (St. Louis, MO).”
- Line 194: please specify the solvent composition ratio DCM:ethanol
Response 6: We apologize for our carelessness. Thank you for your thoughtful suggestion. We have made a further expression. (Line 246).
“The prepared 5 mg of PLGA MPs@KGN (n = 5) were dissolved in component solvent of dichloromethane (DCM) and ethanol (4:1 for volume ratio) of 5.0 mL.”
- Line 189: The "MPs solution" is actually a dispersion. Please correct this.
Response 7: We apologize for our carelessness. Thanks for your thoughtful suggestion. We have corrected it. (Line 238).
“The MPs and MPs@KGN dispersion were sonicated for 2 or 5 min using an ultrasound probe”
- Considering the drug release profiles, is it possible to achieve therapeutic levels of drug release without the application of ultrasound?
Response 8: Yes, your opinion is very rigorous. The effect of KGN on BMMSCs in vitro was dose-dependent. Despite the fact that KGN can be delivered into articular cavities via intraarticular injection, most of the KGN will be absorbed by the circulatory system. In order to maintain a certain concentration of the drug for a certain period of time, more KGN need to be embedded in MPs and more MPs need to be added in the hydrogel, and the toxic side effects of drug burst release may occur. Ultrasound response can achieve on-demand drug release. We have made more detailed description in the part of introduction.
- It is encouraging to see that post hoc statistical analysis have been applied to the results following ANOVA. Please describe these post hoc analysis in the statistical analysis section.
Response 9: Thanks for your thoughtful suggestion. We have made a further expression. (Line 377)
“Statistical Analysis among groups were calculated using a one-way analysis of variance ANOVA after testing for homogeneity of variance (test Levene’s) with post hoc contrasts by Least Significant Difference test (LSD-t).”

Round 2
Reviewer 1 Report
Accepted in present form
Author Response
Thank you very much for your suggestion, which is very helpful to the modification of our article.
Reviewer 3 Report
Accept
Author Response

(The authors gave the same response as above.)

Reviewer 5 Report
The authors' have attended to all the comments and suggestions of the the reviewer. Hence, the manuscript has been improved significantly and can now be considered for publication after minor text editing.
Author Response
Thank you very much for your suggestions. We have further modified the manuscript according to the editor's suggestions.